# Explanatory Factors of Burnout in a Sample of Workers with Disabilities from the Special Employment Centres (SEC) of the Amica Association, Spain

**DOI:** 10.3390/ijerph18095036

**Published:** 2021-05-10

**Authors:** Isabel Gutierrez-Martínez, Josefa González-Santos, Paula Rodríguez-Fernández, Alfredo Jiménez-Eguizábal, Jose Antonio del Barrio-del Campo, Jerónimo J. González-Bernal

**Affiliations:** 1Amica Association of Cantabria, 39300 Torrelavega, Spain; isa.gutier@hotmail.com; 2Department of Health Sciences, University of Burgos, 09001 Burgos, Spain; jejavier@ubu.es; 3Department of Education, University of Burgos, 09001 Burgos, Spain; ajea@ubu.es; 4Department of Education, University of Cantabria, 39005 Santander, Spain; jose.delbarrio@unican.es

**Keywords:** work-related stress, burnout, emotional exhaustion, cynicism, personal efficacy, disabilities, Special Employment Centres SEC

## Abstract

Background: we have based our study on the fact that the labour market is progressively becoming more accessible for people with disabilities. This investigation aims to identify the factors that contribute to high levels of work-related stress in a group of disabled individuals in order to develop policies to prevent it and promote the health of the workforce. Methods: 131 workers from two Special Employment Centres (SECs) of the Amica Association in Cantabria (Spain) participated in the study. Sociodemographic and job-related variables were collected using a questionnaire. Work-related stress was evaluated using the Maslach Burnout Inventory General Survey (MBI-GS), which analyzes emotional exhaustion, cynicism and personal efficacy. Results: the main explanatory factors for higher levels of emotional exhaustion were more than 5 years of service in the company (OR 3.235–IC 95% 1.392–7.519; *p* = 0.006) and bad job satisfaction (OR 7.615–IC 95% 2.467–23.503; *p* = 0.0001); higher levels of cynicism were also explained by bad job satisfaction (OR 8.599–IC 95% 2.481–29.799; *p* = 0.001). Conclusions: future research is needed to facilitate the design of company policies and promote the well-being of the disabled population in the workplace, to avoid pathological conditions such as burnout syndrome.

## 1. Introduction

There are approximately 1 billion people with disabilities worldwide (15%) [1], understanding “disability” as a limitation or inability to participate in main life activities due to physical, sensory, cognitive, intellectual, emotional, psychological or psychiatric problems [2,3]. Unfortunately, this population is submitted to a high stigma [4] originating from an interrelation of components that trigger discriminatory thoughts and behaviours, such as barriers in accessing the working world or poor care quality when accessing health services [5,6,7].

Around 80% of disabled people are of employment age and their incorporation into the working world is an ever-closer reality [1]. Some countries have created laws and clear incentives to promote this inclusion, such as the case of Spain, where individuals with disabilities can develop professionally and choose between sheltered or regular employment [8]. In relation to the latter, the employee joins the community as part of standard companies with conditions that are as similar as possible in terms of work and remuneration to those of workers without disabilities [9]. Regarding protected employments, specific work centres for people with disabilities exist, known as Special Employment Centres (SECs), which are responsible for most of the hiring of this population [10]. These organizations, whose social objective is the inclusion of disabled people, can be built by public administrations or by natural or legal individuals that meet the corresponding civil requirements [11]. Generally, this type of work is low-skilled, with little remuneration and few possibilities to access ordinary employment [12], and linked to the stigma presented by society, it causes a situation of vulnerability that can lead to a risk state [13].

In 1974, Herbert Freudenberg first described a syndrome related to work stress under the term “burnout” [14], which is generally defined by symptoms that appear after the exposure to psychosocial risk factors and that is presented in the form of emotional exhaustion, cynicism and reduced personal efficacy [15,16]. Emotional exhaustion refers to the loss of energy, fatigue, wear and tear that workers can present both physically and cognitively; cynicism means a negative, insensitive or excessively apathetic response to several aspects of the job; and the lack of personal efficacy is a negative feeling towards oneself and the work done [14]. Factors such as the task performed [17], the workspace [17] and the lack of job satisfaction [18] can promote the appearance of burnout, whereas personal strategies such as resilience, self-efficacy, coping with frustration or self-control can reduce the probability of developing this syndrome [19,20].

Work-related stress has been extensively studied in the general population [21,22,23,24,25]. However, it is difficult to find research that addresses the issue in people with disabilities. Motivation and job satisfaction have been shown to predict job success in a sample of employees with intellectual disabilities [26]. Likewise, Flores et al. [27] analysed burnout from a life quality perspective in a group of people with intellectual disabilities, finding higher levels of work-related stress in women.

The integration into the labour market of disabled individuals brings benefits to both society as a whole and to the people with disabilities themselves, since people who have paid employment see their self-esteem and self-perception increased, and they can begin to accomplish personal projects contributing both economically and with their full participation in society [28]. In other words, they get rid of a double stigma: not having a job and having a disability [29,30,31,32]. Nowadays, the integration of people with disabilities in the labour market is becoming a reality, so it is essential to start research in this line in order to establish a profile of protection factors that companies can consult to promote the prevention of work stress. 

Considering this context and based on the data obtained from a sample of workers from the SECs of the Amica Association (Spain), it is proposed to study the factors that contribute to high levels of work stress in a group of disabled people to create policies for preventing it and promoting the health of the workforce.

## 2. Materials and Methods

### 2.1. Study Design—Participants

A cross-sectional study was designed to investigate work-related stress and associated sociodemographic factors in people with disabilities. The studied population consisted of disabled employees of two SECs of the Amica Association in Cantabria, who had been working for more than one month in one of the selected SECs. Those who presented any of the following characteristics during the implementation of the study were excluded: sick leave, retirement or death.

### 2.2. Sample Size

For a population of 1267 employees of the SECs in Cantabria [30], with a margin of error of 9% and a confidence level of 95 %, a minimum sample of 108 subjects is necessary. A total of 131 workers participated in the study, which is considered a representative sample of the studied population, taking into consideration the parameters established in terms of size.

### 2.3. Procedure

The participant selection process was performed by the Human Resources Department of the selected SECs. At first, all active people were included and later, those who did not meet the inclusion criteria were excluded. The application of the evaluation instruments for the collection of the sociodemographic and work-related stress data was performed by the Evaluation Team, except in the case of a worker with a hearing disability, whose application was implemented using sign language and guided by her reference professional. Participation was anonymous and voluntary, so not all of the participating people answered every question.

The signing of a commitment form and a consent report were required, respecting at all times the ethical principles contained in the Helsinki Declaration.

### 2.4. Main Outcomes—Instruments

The main study variable was the work-related stress level, evaluated using the Maslach Burnout Inventory General Survey (MBI-GS) [33], adapted to Spanish and validated by Salanova et al. in the year 2000 [34]. It is a self-administered questionnaire, which consists of 15 items evaluated using a Likert frequency scale that ranges from 1 to 7, distributed in three subscales connected to stress level: emotional exhaustion (5 items), cynicism (4 items) and personal efficacy (6 items). High scores on the emotional exhaustion and cynicism subscales indicate higher work-related stress, but in the case of personal efficacy, are the lower scores which refer worse results. The reliability as internal consistency in each of the subscales is α > 0.70. The administration time is 5 to 10 min.

The secondary variables, considered as possible protective factors against work-related stress after the review of many investigations related to the studied subject, were collected using a questionnaire prepared by the research team [35,36,37]. Sociodemographic data related to gender, age, marital status and educational level were taken into consideration. Categorical variables related to employment were also considered, such as the time worked in the company, type of contract, type of working day and job satisfaction. In addition, although the Amica Association tries to perform a personal itinerary adjusted to the interests, needs and capacities of each employee, it has different centres in which certain tasks are done, for which data about the workplace were collected, depending on whether the employees belonged to the Entorno, Marisma, Horizon or other centres. Other secondary variables were also considered, such as the type of disability, the percentage of disability and the type of support received from the company, whether in the workplace, personnel, follow-up with human resources (HR) or other support.

### 2.5. Statistical Analysis

For the characterization of the sample, the mean and standard deviation (SD) were used in the case of quantitative variables, and the frequency and percentage distribution in qualitative variables. Compliance with the normality criteria of the quantitative variables was assessed using the Kolmogorov–Smirnov test.

Despite the fact that the sample studied did not follow a normal distribution, some authors affirm the possibility of using parametric tests in relation to certain assumptions. Recently, in a study about the robustness of the analysis of variance (ANOVA), the validity of making this analysis in a sample that does not follow a normal distribution was verified [38]. Therefore, although the sample did not present normality, an inferential analysis was performed using the parametric ANOVA test in cases in which there were more than two categories in the independent variables and the non-parametric Mann–Whitney U test when there were two response categories.

All variables with a *p*-value < 0.05 in the univariate analysis were subsequently analysed using binary logistic regressions to determine the explanatory factors for high levels of work-related stress. For this, the dependent variables (emotional exhaustion, cynicism and personal efficacy) were dichotomized according to the mean score of each of them.

For the analysis of statistical significance, a value of *p* < 0.05 was established. Statistical analysis was performed with SPSS version 25 software (IBM-Inc., Chicago, IL, USA).

## 3. Results

A total of 131 people were part of the study sample, with 62% of the participants being men (*n* = 82). Most of the subjects had a medium level disability, between 33% and 65% (*n* = 87), and almost half of the sample had a physical or sensory disability diagnose (47%). The mean score obtained in the primary outcome variable was 40.46 (16.11) (Table 1).

Table 2 shows the differences between groups in the primary outcome variable “emotional exhaustion”. After analysing the emotional exhaustion levels, statistically significant differences were found in relation to the workplace (*p* = 0.001); specifically, the post hoc analysis showed a significantly higher level of emotional exhaustion in those belonging to the Marisma centre compared to those from the Entorno centre (*p* = 0.002). Participants who had been working in the company for a period of time greater than 5 years also reported higher levels of emotional exhaustion compared to those whose service in the SECs did not exceed 5 years (*p* = 0.003), and workers who claimed to have very bad, bad or fair general satisfaction showed to be more emotionally exhausted than the group with good or very good job satisfaction (*p* = 0.001).

When analysing the level of cynicism of the sample according to the different study groups (Table 3), people with more than 5 years of service in the SECs (*p* = 0.005), with a permanent contract (*p* = 0.010), who worked full-time (*p* = 0.05) and with a bad job satisfaction (*p* = 0.007) showed higher levels of cynicism than their counterparts. Taking into account the level of disability, the post hoc analysis showed a significantly higher score in people with a mild disability compared to those with a severe disability, over 65% (*p* = 0.033). Employees at the Marisma centre also reported higher levels of cynicism compared to participants who worked at the Entorno centre (*p* = 0.043).

Regarding personal efficacy, no statistically significant differences were found in the analysis of the differences between groups (Table 4).

The binary logistic regression analysis showed that the explanatory factors for higher levels of emotional exhaustion were more than 5 years of service in the company (OR 3.235–IC 95% 1.392–7.519; *p* = 0.006) and bad job satisfaction (OR 7.615–IC 95% 2.467–23.503; *p* = 0.0001); higher levels of cynicism were mainly explained by bad job satisfaction (OR 8.599–IC 95% 2.481–29.799; *p* = 0.001). The rest of the variables that were significant in the inferential analysis did not demonstrate a significant influence on the primary outcome variables (*p* > 0.05).

## 4. Discussion

It is important to highlight the limited empirical evidence in the scientific literature focused on workers with disabilities. Some authors have analysed psychosocial characteristics such as job satisfaction [39], quality of working life [40,41] and motivation for work in general [11], but burnout, which is one of the most studied aspects related to work fields in the general population, has hardly been considered in people with disabilities, nor have studies been performed comparing the two populations.

Throughout Europe, 79% of workers rank work-related stress as the first health and safety risk [42]. In Spain, the Spanish National Statistics Institute recorded a mean of 4.18 points in the general citizenship (with and without disability of the whole country) and a mean of 4.79 points in Cantabria (specific region in Spain), which is considered a medium stress level on a scale from 1 (not stressful at all) to 7 (very stressful) [43]. In the present research, mean scores similar to those mentioned above were found, so although some participants reported high levels of emotional exhaustion and cynicism and low professional efficacy, this does not indicate the presence of burnout syndrome.

In relation to the influence of gender on burnout, Maslach and Jackson [37] found that women were less likely to develop burnout syndrome, despite having worse levels of exhaustion and professional effectiveness than men [44]. Furthermore, there are more current studies that show higher emotional exhaustion and work-related stress in women [27,45], due to factors such as less autonomy at work, fewer opportunities for development, greater insecurity of job stability and difficulties in reconciling family and work life [46,47]. Although the present research did not find a statistically significant difference in emotional exhaustion, cynicism and professional efficacy scores by gender, women demonstrated slightly higher scores than men on all three subscales.

Considering other relevant sociodemographic variables, García et al. [36] found an inverse relationship between age and work-related stress, probably due to greater experience and better use of coping strategies. Moreover, Angulo et al. [48] showed that living together as a family or partner is a protective factor against work stress due to its connection with the immediate support network, as long as family problems do not occur. Educational level has been shown to be a possible protective factor against work-related stress in the general population [49], but no relevant research has been found for the present analysis that studies burnout according to educational level. In line with these results, our findings show higher levels of work-related stress in people aged 45 years or less, in workers who live alone, and in those with a lower educational level, despite not having a statistically significant difference with their counterparts.

Despite the lack of studies that analyse work-related stress considering the percentage of disability, in the present research the group of people with a disability under 33% showed significantly higher scores in the variable “cynicism” in relation to workers with a severe disability.

An investigation that considered variables related to work such as the time worked in the company or the type of contract showed that people who had been working between 6 and 15 years had a lower work-related stress level than workers with less than 5 years of service [50]. In contrast to these results, people with more than 5 years of service in the CEE showed higher levels of cynicism and emotional exhaustion than those who had been working in the Amica Association centres for a period equal to or under 5 years. Angulo et al. and Leka et al. found a positive correlation between the level of work-related stress and the working day [48,51], probably due to the type of working day itself rather than to the tasks performed [48]. According to these results, full-time workers showed more cynicism than those with a part-time contract in the SECs, which suggests the need to adapt working hours to the personal characteristics of each employee in order to avoid pathological conditions such as burnout syndrome.

One of the main explanatory factors of work stress in the employees of the Amica Association in Cantabria was job satisfaction; people with the worst satisfaction reported significantly higher scores on the subscales of emotional exhaustion and cynicism. Several studies have shown an inverse correlation between burnout syndrome and job satisfaction [52,53,54]. High levels of satisfaction act as a protective factor against the variables that promote the appearance of work-related stress [55]. Motivation and job satisfaction are predictors of job success in the general population [56], and more specifically among people with intellectual disabilities [26]. Going with this idea, Baumgärtner et al. [57] stated that understanding why and how job satisfaction differs for various groups of employees and what can be done to improve the quality of work life in these groups is essential in achieving long-term inclusion of all employees and avoiding stress and burnout during job performance.

This study provides pioneering data on the levels of work-related stress in a group of people with disabilities. However, it should be considered in the context of its strengths and limitations. Despite being a representative sample of the Cantabrian population in terms of numbers, the studied population comes from only two SECs, which could lead to a selection bias. Furthermore, as it is an area of study that is little considered and with a long way to go, relevant sociodemographic and job-related factors that explain the work-related stress of people with disabilities may not have been analysed in this investigation. The lack of previous studies that present this issue makes it difficult to contrast the findings obtained in this research, which has led to the discussion of our results with those of the research performed in the general population. These limitations should be taken into account as they may have influenced the study discoveries and reduced the representativeness of the results. The strengths of this study include the data collection of a large sample of people with disabilities working in two SECs, and the analysis of a wide set of variables. The present investigation starts from a study area little addressed. New information is provided concerning factors that explain the work-related stress of this group, such as the lack of job satisfaction or more than a 5-year service in the SECs. These data should be considered when designing company policies and future research to promote well-being in the workplace and avoid pathological conditions such as burnout syndrome in people with disabilities.

In the growing inclusion of disabled people in society, access to employment with equal rights is necessary to analyse whether this process is being done correctly and promoting health, and to verify that it does not represent an added risk factor. The study of protective factors to promote occupational health of individuals with disabilities should continue, considering disability as a characteristic of the person and not “as a whole”. The typical nature of jobs and participation with rights and responsibilities in society are examples of this new conception, from which similar studies can be developed. Understanding this change of approach promotes progress in aspects that have hardly been taken into account in the population with disabilities, such as work stress. Further research is needed to achieve a real inclusion, where the gaze is on the person.

## 5. Conclusions

Although there were no levels of work stress that could suggest the presence of burnout syndrome, bad job satisfaction and more than a 5-year service in the SECs were the main explanatory factors for the worst scores. For disabled participants under 33%, permanent contract and full-working day also showed significantly worse scores, despite not having influenced the levels of work stress in the binary logistic regression analysis.

## Figures and Tables

**Table 1 ijerph-18-05036-t001:** Descriptive data of the dimensions measured by the MBI-GS.

MBI-GS	Mean	SD	Scoring Interval
Emotional exhaustion	13.64	8.49	5–35
Cynicism	8.72	6.37	4–28
Personal efficacy	29.65	10.73	6–42
Total burnout	40.46	16.11	15–105

MBI-GS: Maslach Burnout Inventory General Survey; SD: standard deviation.

**Table 2 ijerph-18-05036-t002:** Differences between groups in the primary outcome variable “emotional exhaustion” using ANOVA and Mann–Whitney U test.

Sociodemographic Variables	Group	*n*	Mean	SD	*p-*Value
Type of disability	Intellectual	28	13.50	9.343	0.998
	Psychosocial	28	13.29	7.688	
	Physical or sensory	58	13.16	8.203	
	More than one type of disability	10	13.50	6.346	
Percentage of disability	<33%	11	16.45	9.893	0.109
	33–65%	96	14.17	8.713	
	>65%	21	10.48	6.137	
Gender	Male	81	14.19	9.346	0.777
	Female	50	12.76	6.868	
Age	≤45 years	68	14.07	8.686	0.636
	>45 years	62	13.29	8.326	
Marital status	Lives alone	86	12.98	7.828	0.420
	Lives as a couple	45	14.91	9.587	
Educational level	Primary studies	73	12.78	8.294	0.085
	Higher than primary studies	56	14.96	8.714	
Workplace	Entorno	33	9.39	5.238	0.001
	Marisma	53	16.45	8.879	
	Horizon	12	11.08	5.518	
	Other	31	14.71	9.747	
Time worked in the company	≤5 years	51	10.82	6.719	0.003
	>5 years	80	15.44	9.030	
Type of contract	Temporary contract	36	11.17	7.029	0.054
	Permanent contract	95	14.58	8.830	
Type of working day	Full time	94	14.16	8.875	0.194
	Part time	35	12.00	7.178	
Type of support	Support in the position, personnel or follow-up with HR	35	11.89	7.749	0.577
	Other	48	12.58	7.585	
Job satisfaction	Very bad/bad/fair	25	22.76	9.479	0.0001
	Good/very good	106	11.49	6.645	

*n*: number of participants; SD: standard deviation.

**Table 3 ijerph-18-05036-t003:** Differences between groups in the primary outcome variable “cynicism” using ANOVA and Mann–Whitney U test.

Sociodemographic Variables	Group	*n*	Mean	SD	*p-*Value
Type of disability	Intellectual	28	8.86	6.234	0.557
	Psychosocial	27	8.89	7.628	
	Physical or sensory	57	8.61	5.882	
	More than one type of disability	10	5.80	3.360	
Percentage of disability	<33%	11	12.55	5.484	0.033
	33–65%	95	8.66	6.558	
	>65%	21	6.38	4.995	
Gender	Male	82	8.67	6.761	0.418
	Female	48	8.81	5.697	
Age	≤45 years	67	9.21	6.988	0.887
	>45 years	62	8.27	5.660	
Marital status	Lives alone	86	8.78	6.689	0.780
	Lives as a couple	44	8.61	5.756	
Educational level	Primary studies	74	7.86	5.392	0.107
	Higher than primary studies	54	9.85	7.497	
Workplace	Entorno	33	6.48	4.711	0.034
	Marisma	52	10.48	7.312	
	Horizon	12	8.75	4.288	
	Other	31	7.97	6.102	
Time worked in the company	≤5 years	51	6.78	4.397	0.005
	>5 years	79	9.97	7.113	
Type of contract	Temporary contract	36	6.69	4.689	0.010
	Permanent contract	94	9.50	6.763	
Type of working day	Full time	93	9.38	6.773	0.050
	Part time	35	6.97	4.884	
Type of support	Support in the position, personnel or follow-up with HR	36	8.19	6.815	0.685
	Other	48	8.42	5.870	
Job satisfaction	Very bad/bad/fair	24	12.50	8.547	0.007
	Good/very good	106	7.87	5.458	

*n*: number of participants; SD: standard deviation.

**Table 4 ijerph-18-05036-t004:** Differences between groups in the primary outcome variable “personal efficacy” using ANOVA and Mann–Whitney U test.

Sociodemographic Variables	Group	*n*	Mean	SD	*p-*Value
Type of disability	Intellectual	28	28.21	10.727	0.102
	Psychosocial	28	31.07	9.764	
	Physical or sensory	57	28.09	11.645	
	More than one type of disability	10	36.50	5.148	
Percentage of disability	<33%	10	28.10	9.024	0.896
	33–65%	97	29.49	11.201	
	>65%	21	30.05	9.811	
Gender	Male	82	29.95	9.970	0.849
	Female	49	29.14	11.90	
Age	≤45 years	69	30.32	9.990	0.806
	>45 years	61	29.03	11.587	
Marital status	Lives alone	87	29.59	10.407	0.669
	Lives as a couple	44	29.77	11.469	
Educational level	Primary studies	74	29.28	10.573	0.532
	Higher than primary studies	55	29.75	11.022	
Workplace	Entorno	32	31.50	8.948	0.032
	Marisma	54	31.54	10.302	
	Horizon	12	26.00	11.855	
	Other	31	25.45	11.772	
Time worked in the company	≤5 years	50	30.42	11.546	0.213
	>5 years	81	29.17	10.242	
Type of contract	Temporary contract	36	30.25	12.122	0.334
	Permanent contract	95	29.42	10.216	
Type of working day	Full time	94	29.01	10.766	0.211
	Part time	35	31.23	10.855	
Type of support	Support in the position, personnel or follow-up with HR	35	28.69	11.463	0.548
	Other	48	30.56	10.177	
Job satisfaction	Very bad/bad/fair	25	32.12	8.506	0.260
	Good/very good	106	29.07	11.147	

*n*: number of participants; SD: standard deviation.

## Data Availability

Not applicable.

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
