# Peer review of "Explanatory Factors of Burnout in a Sample of Workers with Disabilities from the Special Employment Centres (SEC) of the Amica Association, Spain"

_ijerph, 2021, doi:10.3390/ijerph18095036_

Round 1
Reviewer 1 Report
- Please consider reviewing grammar and syntax throughout the paper
- The paper compares difference in workplaces in the result section, but these are not mentioned in the introduction or methods. Kindly consider giving a context to these results.
- Discussion section can be improved. Consider adding a paragraph on the implications of the results, the possible stays that have or can be taken to address the issue in the relevant institutions, and provide specific guidance for future research.
Author Response
- Response to Reviewer 1:
First of all, we would like to express our sincere gratitude for all comments and suggestions received from the Reviewer 1. This information has certainly enriched the text for its best understanding, thank you very much indeed. We have clarified the reviewer1’s questions. We have introduced the required changes both in our answers to the specific comments and in the final manuscript V2.
Broad comments:
Please consider reviewing grammar and syntax throughout the paper.
Response: Thank you very much for pointing it out. Grammar and syntax have been reviewed by an expert.
The paper compares difference in workplaces in the result section, but these are not mentioned in the introduction or methods. Kindly consider giving a context to these results.
Response: Thank you very much for pointing it out. We have added information related to the workplace in the methods section.
“…. In addition, although the Amica Association tries to carry out a personal itinerary adjusted to the interests, needs and capacities of each employee, it has different centers in which certain tasks are carried out, for which data about the workplace were collected, depending on whether the employees belonged to the Entorno, Marisma, Horizon, or other centers.”
Discussion section can be improved. Consider adding a paragraph on the implications of the results, the possible stays that have or can be taken to address the issue in the relevant institutions and provide specific guidance for future research.
Response: Thank you very much for pointing it out. We have added a paragraph on the implications of the results and future research.
“The present investigation starts from a study area little addressed… In the growing inclusion of people with disabilities in society, access to employment with equal rights is included and it is necessary to analyze whether this process is being carried out correctly, promoting health and verifying that it does not rep-resent an added risk factor. The study of protective factors to promote occupational health of people with disabilities should continue, considering disability as a characteristic of the person and not “as a whole”. The typical nature of jobs and participation with rights and responsibilities in society are examples of this new conception, from which similar studies can be developed. Understanding this change of approach promotes progress in aspects that have hardly been taken into account in the population with disabilities, such as work stress. Further research is needed to achieve a real inclusion, where the gaze is on the person.”
Thank you very much.
Josefa González Santos
Reviewer 2 Report
Dear Authors, this paper is distant from novelty polarly, yet, written correctly.
Author Response
Response to Reviewer 2:
First of all, we would like to express our sincere thanks for the favourable feedback received from Reviewer 2, thank you very much indeed. For us it is an honour that a reviewer of a journal of such impact has this opinion of our research.
Thank you very much.
Josefa González Santos
Reviewer 3 Report
The article is structurally correct and the subject is of interest. Therefore, its publication is recommended. The authors would only be asked for a conclusion that provides more information on how the present study can contribute to the improvement of the situation of workers and what proposal the authors of the work have to introduce the data of this study into social and health policies . It would also be interesting to introduce factors that will help to understand the data disaggregated by gender, as indicated but not developed.
Author Response
Response to Reviewer 3:
First of all, we would like to express our sincere gratitude for all comments and suggestions received from the Reviewer 3. This information has certainly enriched the text for its best understanding, thank you very much indeed. We have clarified the reviewer3’s questions. We have introduced the required changes both in our answers to the specific comments and in the final manuscript V2.
Broad comments:
The article is structurally correct and the subject is of interest. Therefore, its publication is recommended. The authors would only be asked for a conclusion that provides more information on how the present study can contribute to the improvement of the situation of workers and what proposal the authors of the work have to introduce the data of this study into social and health policies
Response: Thank you very much for pointing it out. We have added a paragraph on the implications of the results and future research.
“The present investigation starts from a study area little addressed… In the growing inclusion of people with disabilities in society, access to employment with equal rights is included and it is necessary to analyze whether this process is being carried out correctly, promoting health and verifying that it does not rep-resent an added risk factor. The study of protective factors to promote occupational health of people with disabilities should continue, considering disability as a characteristic of the person and not “as a whole”. The typical nature of jobs and participation with rights and responsibilities in society are examples of this new conception, from which similar studies can be developed. Understanding this change of approach promotes progress in aspects that have hardly been taken into account in the population with disabilities, such as work stress. Further research is needed to achieve a real inclusion, where the gaze is on the person.”
It would also be interesting to introduce factors that will help to understand the data disaggregated by gender, as indicated but not developed.
Response: Thank you very much for pointing it out. We have added information related to the influence of gender on burnout of people with disabilities in the introduction:
“Likewise, Flores et al. [27] analyzed burnout from a quality of life perspective in a sample of people with intellectual disabilities, find-ing higher levels of work-related stress in women.”
Thank you very much.
Josefa González Santos
Reviewer 4 Report
This study examines the relationship of burnout symptoms with a number of demographic and job-related variables among a sample of workers with disabilities in special employment centers. This is an interesting and relevant topic for study, as little is known about the work-related wellbeing of individuals with a disability. Moreover, if individuals with disabilities experience high levels burnout this may preclude prolonged labor participation.
Nevertheless, I have a number of concerns with the present manuscript. In the first place, no theoretical perspective on burnout is used in this study. As a result, the studied correlates of burnout are limited to a number of demographic variables and characteristics of the job contract, including tenure, number of working hours, and contract type. None of the theoretically important job demands (e.g., work load, physical demands, emotional demands, etc.) or job resources (social support, autonomy, feedback, etc.) are included in the study. Job satisfaction is included as a correlate of burnout, but this should probably be seen as another outcome variable rather than a predictor of burnout.
Secondly, the introduction focuses solely on labor participation of workers with a disability. The research question is not adequately introduced, and relevant literature on burnout among workers with a disability is not mentioned. This literature is partly mentioned in the Discussion section, but there it seems misplaced, as it goes into detail on subjects that are irrelevant given the results of the study (e.g., influence of gender on burnout).
In the third place, the main statistical analyses are not suitable for the research question. A logistic regression is carried out on burnout variables that are dichotomized at the mean score. This procedure strongly lowers power. I would advocate linear regression (with dummy variables for the predictors), using maximum likelihood estimation and robust standard errors to account for the nonnormal distribution of the outcome variables. A logistic regression would only be justified when the official cut-off scores are used for high risk of burnout (although I would prefer linear regression).
Finally, there were many typos, and some sentences that were incomprehensible (e.g., page 6, line 182-191). Proof reading by a native English speaker would be useful.
More detailed comments
- The description of variables is unclear. Especially the meaning of the variable type of support received from the company remains unclear.
- According to paragraph 2.1, the study population consists of participants with a disability equal to or greater than 33%. However, in the study there are 11 participants with a disability percentage lower than 33%. This is inconsistent. Moreover, it is not fully clear what this disability percentage refers to. Please explain.
- If I understand correctly, 1267 potential participants were invited for this study, and 131 actually participated (10% response rate). This low response rate does not justify the claim that the sample is representative for the population.
- Was the study approved by an ethical review board?
Author Response
Response to Reviewer 4:
First of all, we would like to express our sincere gratitude for all comments and suggestions received from the Reviewer 4. This information has certainly enriched the text for its best understanding, thank you very much indeed. We have clarified the reviewer4’s questions. We have introduced the required changes both in our answers to the specific comments and in the final manuscript V2.
Broad comments:
Nevertheless, I have a number of concerns with the present manuscript. In the first place, no theoretical perspective on burnout is used in this study. As a result, the studied correlates of burnout are limited to a number of demographic variables and characteristics of the job contract, including tenure, number of working hours, and contract type. None of the theoretically important job demands (e.g., work load, physical demands, emotional demands, etc.) or job resources (social support, autonomy, feedback, etc.) are included in the study. Job satisfaction is included as a correlate of burnout, but this should probably be seen as another outcome variable rather than a predictor of burnout.
Response: Thank you very much for pointing it out. The absence of studies investigating burnout in people with disabilities made it difficult to determine the secondary study variables. Taking into account previous research in other populations, the secondary study variables were determined, also considering the characteristics of the sample. The lack of adapted and validated instruments in the population with disabilities (physical, sensory, cognitive, intellectual, etc.) made it impossible to analyze relevant variables in this field of study, such as workload. All these limitations appear in the manuscript. Labor resources were evaluated using the variable “type of support” and in relation to job satisfaction, it will be studied further in future research
Secondly, the introduction focuses solely on labor participation of workers with a disability. The research question is not adequately introduced, and relevant literature on burnout among workers with a disability is not mentioned. This literature is partly mentioned in the Discussion section, but there it seems misplaced, as it goes into detail on subjects that are irrelevant given the results of the study (e.g., influence of gender on burnout).
Response: Thank you very much for pointing it out. The lack of studies related to burnout in employees with disabilities makes it difficult to introduce the subject of study using the relevant literature on burnout among workers with disabilities. The literature mentioned by the editor in the introduction section has been introduced as follows:
“Work-related stress has been extensively studied in the general population [21–25], however, it is difficult to find research that addresses the issue in people with disabilities. Motivation and job satisfaction have been shown to predict job success in a sample of em-ployees with intellectual disabilities [26]. Likewise, Flores et al. [27] analyzed burnout from a quality of life perspective in a sample of people with intellectual disabilities, find-ing higher levels of work-related stress in women.”
In the third place, the main statistical analyses are not suitable for the research question. A logistic regression is carried out on burnout variables that are dichotomized at the mean score. This procedure strongly lowers power. I would advocate linear regression (with dummy variables for the predictors), using maximum likelihood estimation and robust standard errors to account for the nonnormal distribution of the outcome variables. A logistic regression would only be justified when the official cut-off scores are used for high risk of burnout (although I would prefer linear regression).
Response: Thank you very much for pointing it out. As the necessary assumptions to create a linear regression model were not met (linearity, independence, homoscedasticity, normality, non-collinearity), it was decided to use a logistic regression model. If the assumptions had been met, a linear regression analysis would have been performed.
Finally, there were many typos, and some sentences that were incomprehensible (e.g., page 6, line 182-191). Proof reading by a native English speaker would be useful.
Response: Thank you very much for pointing it out. We have made the required changes as follows:
“79% of Europeans place work-related in the first place in terms of health and safety risks [42], and the Spanish National Statistics Institute recorded a mean of 4.18 points in the general population (with and without disability) in the country as a whole, and a mean of 4.79 points in Cantabria, which is considered a medium stress level on a scale from 1 (not at all stressful) to 7 (very stressful) [43].”
The description of variables is unclear. Especially the meaning of the variable type of support received from the company remains unclear.
Response: Thank you very much for pointing it out. We have clarified the description of variables.
“…Sociodemographic data related to gender, age, marital status and educational level were collected. Categorical variables related to employment were also considered, such as the time worked in the company, type of contract, type of working day and job satisfaction. In addition, although the Amica Association tries to carry out a personal itinerary adjusted to the interests, needs and capacities of each employee, it has different centers in which certain tasks are carried out, for which data about the workplace were collected, depending on whether the employees belonged to the Entorno, Marisma, Horizon, or other centers. Other secondary variables were also considered, such as the type of disability, the percentage of disability and the type of support received from the company, whether in the workplace, personnel, follow-up with human resources (HR) or other support.”
According to paragraph 2.1, the study population consists of participants with a disability equal to or greater than 33%. However, in the study there are 11 participants with a disability percentage lower than 33%. This is inconsistent. Moreover, it is not fully clear what this disability percentage refers to.
Response: Thank you very much for pointing it out. There was an error in the inclusion criteria and we have modified it.
“The study population consisted of employees with disabilities of two SEC of the Amica Association in Cantabria, who had been working for more than one month in one of the selected SEC.”
If I understand correctly, 1267 potential participants were invited for this study, and 131 actually participated (10% response rate). This low response rate does not justify the claim that the sample is representative for the population.
Response: Thank you very much for pointing it out. 1267 employees with disabilities work in the SEC of the Amica Association. Considering a margin of error of 9% and a confidence level of 95%, to obtain a representative sample a minimum of 108 subjects are needed. Due to the situation caused by the State of Alarm derived from the COVID-19 pandemic, activated on March 14 of 2020, the application of the evaluation instruments was suspended when 132 people had been evaluated. It was decided to carry out the study with this sample, representative in terms of size for the population studied, since it was considered that the data after the State of Alarm could be distorted due to the exceptional situation they experienced and the implicit stress in the quarantine period.
Was the study approved by an ethical review board?
Response: Thank you very much for pointing it out. The study was approved by the center review board (Amica 2020).
Thank you very much.
Josefa González Santos
Reviewer 5 Report
First of all, we would like to thank them for dealing with this type of population so in need of labor integration policies and strategies. I believe that the subject of your study is clearly defined in terms of the population studied, the risk factors analyzed and the expected results.
The individuals selected for the study have special characteristics that minimize any selection bias and make the study more attractive.
The results have been measured accurately and data on the reliability of the scales are provided to minimize possible measurement biases. In addition, the instrument used is commonly used to assess the construct of burnout syndrome.
However, with respect to the results, I would like to make a few comments for your consideration, as I believe that these could further improve the communication of the results obtained. (1) They do not provide data on the total score of the burnout scale and I believe that doing so would be essential for several reasons: (a) it would allow us to know whether or not the incidence of the syndrome in this type of population differs from the incidence rates recorded in the general population, and (b) it would facilitate the interpretation of the relationships between the different dimensions or subscales of the syndrome. In any case, if they decide not to include these data, I believe it is necessary to explain the reasons for such a decision.
On the other hand, they use odds ratios to report the association between sociodemographic factors and the dimensions of the syndrome, but I understand that given that these factors have different levels, it would be more correct to report this relationship using the RR (relative risk).
Another aspect to consider, although they report the differences between groups in the dimensions of the syndrome, they do not provide information on the size of the effect, so it is not possible to assess the true significance of these differences.
Finally, I hope you understand that these objections are formulated with the sole purpose of improving the communication of your results, as I have a very good opinion of your work, since it is well grounded theoretically and may be of interest to the readers of this journal. In addition, the results may be of great interest for the design of labor integration policies for the population studied.
Author Response
Response to Reviewer 5:
First of all, we would like to express our sincere gratitude for all comments and suggestions received from the Reviewer 5. This information has certainly enriched the text for its best understanding, thank you very much indeed. We have clarified the reviewer5’s questions. We have introduced the required changes both in our answers to the specific comments and in the final manuscript V2.
Broad comments:
However, with respect to the results, I would like to make a few comments for your consideration, as I believe that these could further improve the communication of the results obtained. (1) They do not provide data on the total score of the burnout scale and I believe that doing so would be essential for several reasons: (a) it would allow us to know whether or not the incidence of the syndrome in this type of population differs from the incidence rates recorded in the general population, and (b) it would facilitate the interpretation of the relationships between the different dimensions or subscales of the syndrome. In any case, if they decide not to include these data, I believe it is necessary to explain the reasons for such a decision.
Response: Thank you very much for pointing it out. Data have been provided on the total score of the burnout scale as follows
“The mean score obtained in the primary outcome variable was 40.46 (16.11) (Table 1).”
Table 1. Descriptive data of the dimensions measured by the MBI-GS.
|
|
Mean |
SD |
Scoring interval |
|
Emotional exhaustion |
13.64 |
8.49 |
5- 35 |
|
Cynicism |
8.72 |
6.37 |
4- 28 |
|
Personal efficacy |
29.65 |
10.73 |
6- 42 |
|
Total burnout |
40.46 |
16.11 |
15- 105 |
MBI-GS: Maslach Burnout Inventory General Survey; SD: Standard deviation.
On the other hand, they use odds ratios to report the association between sociodemographic factors and the dimensions of the syndrome, but I understand that given that these factors have different levels, it would be more correct to report this relationship using the RR (relative risk).
Response: Thank you very much for pointing it out. Although it could have been expressed in terms of RR, it was also intended to investigate the effect of other variables on the relationship between two binary variables through logistic regression, for this reason OR were used.
Another aspect to consider, although they report the differences between groups in the dimensions of the syndrome, they do not provide information on the size of the effect, so it is not possible to assess the true significance of these differences.
Response: Thank you very much for pointing it out. The estimate of the effect size and its confidence intervals can be estimated imprecisely in the face of violations of the assumptions of normality and homoscedasticity. These inaccuracies can therefore lead to substantive errors in the interpretation of the data, so it was decided not to include information about the size of the effect.
Grissom, R.J.; Kim, J.J. Review of assumptions and problems in the appropriate conceptualization of effect size. Psychol. Methods 2001, 6, 135–146. Doi: 10.1037/1082-989X.6.2.135
Thank you very much.
Josefa González Santos
Round 2
Reviewer 4 Report
I do think the manuscript has improved, especially with regard to the English writing, and I want to thank the authors for their reply to my comments.
Unfortunately, however, I still believe the study has limited theoretical and practical value due to the selection of correlates of burnout. Moreover, the main message of the article remains somewhat unclear. E.g., why is it important to show that individuals with a disability have a higher risk of burnout when they work in a certain workplace but not in another? What can be learned from this result? Finally, I still do not agree with the analytic strategy with dichotomization at the mean for the burnout measures.